# Effects of Intracerebral Aminophylline Dosing on Catalepsy and Gait in an Animal Model of Parkinson’s Disease

**DOI:** 10.3390/ijms25105191

**Published:** 2024-05-10

**Authors:** Érica de Moraes Santos Corrêa, Gustavo Christofoletti, Albert Schiaveto de Souza

**Affiliations:** Faculty of Medicine, Institute of Health, Federal University of Mato Grosso do Sul, UFMS, Campo Grande 79060-900, Brazil; ericamscorrea@gmail.com (É.d.M.S.C.); g.christofoletti@ufms.br (G.C.)

**Keywords:** Parkinson’s disease, dopaminergic receptors, adenosinergic receptors, catalepsy, gait, haloperidol-induced model

## Abstract

Parkinson’s disease (PD) is a progressive disorder characterized by the apoptosis of dopaminergic neurons in the basal ganglia. This study explored the potential effects of aminophylline, a non-selective adenosine A_1_ and A_2A_ receptor antagonist, on catalepsy and gait in a haloperidol-induced PD model. Sixty adult male Swiss mice were surgically implanted with guide cannulas that targeted the basal ganglia. After seven days, the mice received intraperitoneal injections of either haloperidol (experimental group, PD-induced model) or saline solution (control group, non-PD-induced model), followed by intracerebral infusions of aminophylline. The assessments included catalepsy testing on the bar and gait analysis using the Open Field Maze. A two-way repeated-measures analysis of variance (ANOVA), followed by Tukey’s post hoc tests, was employed to evaluate the impact of groups (experimental × control), aminophylline (60 nM × 120 nM × saline/placebo), and interactions. Significance was set at 5%. The results revealed that the systemic administration of haloperidol in the experimental group increased catalepsy and dysfunction of gait that paralleled the observations in PD. Co-treatment with aminophylline at 60 nM and 120 nM reversed catalepsy in the experimental group but did not restore the normal gait pattern of the animals. In the non-PD induced group, which did not present any signs of catalepsy or motor dysfunctions, the intracerebral dose of aminophylline did not exert any interference on reaction time for catalepsy but increased walking distance in the Open Field Maze. Considering the results, this study highlights important adenosine interactions in the basal ganglia of animals with and without signs comparable to those of PD. These findings offer valuable insights into the neurobiology of PD and emphasize the importance of exploring novel therapeutic strategies to improve patient’s catalepsy and gait.

## 1. Introduction

Parkinson’s disease (PD) is a chronic condition that affects 100–200 individuals per 100,000 inhabitants. The disease can be diagnosed in younger adults, but its incidence is higher in advanced ages [1,2]. It is one of the most common movement disorders affecting the elderly and the second most prevalent neurodegenerative disease, surpassed only by Alzheimer’s disease [3,4].

PD is associated with motor dysfunction. Patients with PD often present with bradykinesia, muscle rigidity, freezing, and gait disturbances [4,5]. The loss of automaticity is a key deficit in PD. As the disease progresses, patients experience difficulties in performing simultaneous tasks, making ordinary activities a real challenge [6]. 

The physiopathology of PD involves progressive apoptosis in various brain structures, with an emphasis on dopaminergic neurons in the basal ganglia. The degeneration of dopaminergic neurons leads to deficits in voluntary movements. In addition to dopamine, other neurotransmitters are affected by PD. The intrinsic mechanisms underlying PD have not yet been fully elucidated. However, the literature indicates a potential effect of acetylcholine, serotonin, and adenosine on the patients’ motor functions [7,8,9].

Levodopa is the primary therapeutic agent used for the treatment of PD patients. It is highly effective in the early stages of the disease, but motor dysfunction is present in most patients after a long period of continuous administration [10,11,12]. Considering the challenges related to the side effects of levodopa, it is important to study new pharmacological therapies to improve the life expectancy of PD patients.

New drugs that act as anti-PD agents have been investigated with promising results [13,14,15]. The literature indicates some substances with therapeutic potential, including those acting on adenosinergic, glutamatergic, cannabinoid, opioid, α2-adrenergic, nicotinic, muscarinic, and cholinergic receptors. As numerous pathways are affected in PD, studies have focused on both dopaminergic and non-dopaminergic systems [16,17]. 

Adenosine is one of the substances that influence the basal ganglia and may exert an effect on PD patients. The pioneering work of Fuxe and Ungerstedt demonstrated the modulatory effects of adenosine on the dopaminergic system, highlighting the existence of antagonistic interactions [18]. Adenosine acts via purinergic receptors, which are divided into four distinct subtypes: A_1_, A_2A_, A_2B,_ and A_3_ [19]. A_2A_ receptors are currently considered a therapeutic target in PD because they have a compensatory effect on dopamine deficiency [20]. 

Previous studies have confirmed that some substances derived from methylxanthines can act on the adenosinergic system, reducing oxidative stress and inflammation in the nervous system [21,22]. Methylxanthines are substances formed by the methylation of xanthines, such as caffeine and theophylline. The anti-PD properties of methylxanthines occur through the blocking of striatal adenosine receptors [23,24]. Adenosine appears to provide an inhibitory tone to various brain regions, and the stimulation of motor behaviors promoted by caffeine or theophylline is attributed to the non-selective antagonism of A_1_ and A_2A_ receptors [25].

Both A_1_ and A_2A_ adenosinergic receptors are known to modulate the dopaminergic system. A_1_ receptors are distributed throughout the brain and regulate neurotransmitter release and neuronal firing. A_2A_ receptors are expressed in regions rich in dopamine, reducing the activity of D_2_ receptors on dopaminergic agonists [26,27,28,29]. Adenosine receptors, therefore, play an important role in motor control. Activation of adenosine A_2A_ receptors inhibits effects mediated by dopamine D_2_ receptors, likely through direct interactions between these two structures [30,31]. 

The literature shows that the antagonism exerted by methylxanthines has stood out for its therapeutic potential not only because of its anti-PD effects on adenosine A_2A_ receptors but also because of its ability to modulate neuronal and glial functions, producing neuroprotective effects [31,32,33]. Despite all scientific advances, the literature highlights the complexity of the therapeutic use of adenosine agents in PD and the need for further investigation. Current studies have shown important findings, but some often lack consistency in their results. To provide valuable insights into the intricate neurobiology of PD, we investigated the therapeutic effects of aminophylline, a nonselective adenosine A_1_ and A_2A_ receptor antagonist, on catalepsy and gait in an animal model of PD. 

To better highlight the findings, we divided the manuscript into Introduction, Results, Discussion, Materials and Methods, and Conclusion sections. With this design, we believe the reader can understand key results and their respective discussions before reviewing the materials and methods used.

## 2. Results

### 2.1. Results of the Catalepsy Test

Sixty male Swiss mice were divided into experimental and control groups. Animals in the experimental group, which were subjected to haloperidol treatment, exhibited catalepsy signs similar to those observed in patients with PD. Animals in the control group, which were subjected to saline injection, did not show signs of catalepsy. Animals in the experimental group treated with saline solution experienced worsening catalepsy. Animals in the experimental group treated with aminophylline showed a significant improvement in catalepsy. Animals in the control group showed slight fluctuations in the reaction time, but the values were significantly lower than those in the experimental group. 

Two-way repeated-measures ANOVA confirmed a main effect of group (*p* < 0.001), but no effect of time (*p* = 0.209) or group × time interaction (*p* = 0.831). This means that, when comparing all six subgroups, the main difference in catalepsy involved groups with or without haloperidol. However, Tukey’s post hoc test confirmed the benefits of aminophylline treatment in the experimental group, both at 60 nM and 120 nM (*p* < 0.05). The findings indicated an improvement in the haloperidol-induced signs of catalepsy when comparing the results of group 1 (without aminophylline) with those of groups 2 and 3 (aminophylline at 60 and 120 nM, respectively). Figure 1 illustrates the catalepsy reaction time for the groups.

### 2.2. Results of the Open Field Maze

Figure 2 shows the walking pattern (number of quadrants crossed) and motor behavior (vertical exploration) of the animals. Animals in the experimental groups (G1, G2, and G3) walked fewer quadrants than those in the control groups (G4, G5, and G6). This finding indicates that neither dose of aminophylline could reverse gait dysfunction in the experimental group. Animals in group 5 (the group without haloperidol but with aminophylline at 60 nM) walked more quadrants than animals in the other five groups (*p* < 0.05). This result shows that 60 nM is better than no aminophylline or aminophylline at 120 nM for the walking pattern of animals without PD-induced signs. 

In terms of vertical exploration, the animals in the experimental groups (G1, G2, and G3) showed worse climbing results than those in the control groups (G4, G5, and G6). This finding indicates that aminophylline did not improve vertical exploration in PD-induced mice. The best result for vertical exploration occurred in group 5, which was characterized by not receiving haloperidol and having aminophylline at 60 nM (*p* < 0.05 when comparing all the subgroups). Aminophylline at 120 nM in animals in the control group (group 6) did not show any benefits compared to those in group 4 (control group without aminophylline).

## 3. Discussion

In this study, we observed that a selective antagonist of D_2_ dopamine receptors, haloperidol, induced catalepsy and motor dysfunctions similar to those observed in patients with PD. Catalepsy was evaluated using the bar test and was completely reversed by aminophylline at 60 nM and 120 nM. In contrast, as evidenced by the Open Field Maze test, the walking and climbing patterns were not reversed by aminophylline. In animals of the control group (without haloperidol), aminophylline at 60 nM demonstrated a stimulating effect on motor activity. Understanding these parameters is important to elucidate the motor signs in patients with PD and to propose new therapies.

Levodopa is the most effective treatment for PD. However, because it is not effective for all patients in later stages of the disease, new approaches have been proposed [34,35,36,37,38]. The development of new therapies aims to prevent the adverse effects of prolonged levodopa use, as well as to provide protection and slow the progression of PD [39,40,41].

The use of haloperidol resulted in catalepsy and walking dysfunction. These findings are consistent with those of previous studies that observed an increase in bar catalepsy time in groups treated with haloperidol [42,43]. The results of our study corroborate the previous findings that have associated the inhibition of dopamine receptors in the basal ganglia with catalepsy and hypokinesia patterns in animals with symptoms similar to those observed in PD [34,44,45].

Previous studies on adenosine and dopamine have shown that alterations of these receptors in the nigrostriatal pathway may explain the pathogenic basis of extrapyramidal disorders, including PD [46,47]. Adenosine acts as a key neuromodulator in the basal ganglia and plays a crucial role in motor function [48]. The literature indicates that the blockade of A_2A_ receptors can stimulate effects mediated by dopamine D_2_ receptors, probably through direct interactions between these two structures. Adenosine receptors play an important role in modulating dopaminergic neurotransmission. This is because of the strategic localization of different subtypes of adenosine and dopamine receptors in the striatonigral and striatopallidal pathways [49,50,51].

Ferré et al. [29] suggested that interactions between the adenosinergic and dopaminergic systems can be explained by the predominance of the striatal heteromeric receptors A_2A_ and D_2_. A_2A_-D_2_ receptors have a tetrameric structure composed of two homodimers, allowing multiple allosteric interactions between different ligands, both agonists and antagonists. This model may elucidate most of the effects of A_2A_ and D_2_ ligands, including the psychostimulant effects of caffeine and theophylline.

Mandhane et al. [52] induced catalepsy by using haloperidol and investigated the anti-cataleptic effects of adenosine. Pretreatment of animals with theophylline, a non-selective adenosine receptor antagonist, or 3,7-dimethyl-1-propargylxanthine (DMPX), a selective adenosine A_2_ receptor antagonist, significantly reversed haloperidol-induced catalepsy. This finding is consistent with that of our study, in which aminophylline completely reversed the cataleptic state in animals that received an intraperitoneal injection of haloperidol.

A complementary study by González-Lugo et al. [42] further supports our findings on the effects of aminophylline. Haloperidol-induced catalepsy was reversed by the systemic administration of theophylline in combination with benztropine, an anticholinergic antagonist. This caused a reduction in catalepsy intensity, which was evaluated by using the bar catalepsy test. In our study, exclusive administration of aminophylline completely reversed catalepsy, even without an anticholinergic drug. This result suggests greater efficacy of aminophylline, possibly owing to its intracerebral route of administration.

Trevitt et al. [53] conducted two experiments to investigate the effects of adenosine antagonists in PD-induced mice. In the first experiment, haloperidol was administered to induce catalepsy. Treatment with caffeine, CPT (selective adenosine A_1_ receptor antagonist), and SCH58261 (selective adenosine A_2A_ receptor antagonist) significantly reduced catalepsy. In the second experiment, haloperidol was used to suppress locomotor activity, and treatment with caffeine increased locomotion but not at all doses. Our results differ from those of Trevitt et al. [53], as aminophylline did not reverse the motor dysfunction induced by haloperidol. The accumulated knowledge about the biochemical properties of A_2A_ and D_2_ receptors offers new therapeutic possibilities for PD and other neurological disorders of the basal ganglia. 

Interestingly, aminophylline yielded varied outcomes depending on the dosage administered. This can be explained by substances that induce modifications in a biphasic manner, which can either stimulate or inhibit motor functions [54]. The results of the present study suggest that aminophylline possesses biphasic effects, stimulating motor functions, especially at 60 nM.

### Limitation

This study had some limitations. First, as this is one of the first attempts to investigate aminophylline in an animal model of PD, we could not completely understand why aminophylline did not improve vertical exploration. Further studies are required to explore this issue. Second, we acknowledge that the animals should have been evaluated before and after the surgical procedures. We opted to assess the animals only at a later time point because several studies presented this methodological design, possibly to avoid learning effects on the animals in the catalepsy test and the Open Field Maze. We encourage further studies to address pre- and post-assessments in animals with and without aminophylline. Finally, the results observed in animals may not directly translate to those observed in patients with PD. While the findings are encouraging, further studies should investigate the effects of aminophylline (or adenosine receptor antagonists, such as theophylline) in patients with PD.

## 4. Materials and Methods

Sixty adult Swiss mice, weighing 30–40 g, were used in this study. The animals were housed in an animal facility with ad libitum access to food and water until the beginning of the experiment. The light cycle (12/12 h, lights on at 6:00 a.m. and off at 6:00 p.m.) and the environmental temperature (23.0 ± 1.0 °C) were controlled. This study was approved by the Institutional Ethics Committee. All experiments complied with regulations and guidelines related to animal welfare.

The animals were divided into two groups. The experimental group consisted of 30 animals that were subjected to intraperitoneal administration of haloperidol (haloperidol from Janssen Cilag). As haloperidol is a dopamine D_2_-receptor antagonist, this group started to present with catalepsy and motor dysfunction, which are common symptoms of PD [53]. The control group consisted of 30 animals that were administered saline solution and did not present any PD symptoms. Next, animals from both groups were subjected to stereotaxic surgery to investigate the effects of aminophylline (at 60 nM and 120 nM) on catalepsy and gait [54]. Aminophylline (aminophylline was from Research Biochemicals International) is an antagonist of A_1_ and A_2A_ adenosine receptors, and, with this methodological design, it was possible to investigate its effects in animal models simulating PD and in control animals without PD symptoms. A third subgroup of saline solutions was used to provide placebo comparisons with the aminophylline subgroups. Figure 3 details the distribution of animals in each group. Figure 4 (and Appendix A) shows the surgical procedures adopted in this study.

### 4.1. Stereotaxic Surgery and Intracerebral Drug Administration

All animals underwent stereotaxic surgery for bilateral implant stain-guided cannulas. The mice were anesthetized with ketamine (100 mg/kg) and xylazine (10 mg/kg) at a volume of 10 mL/kg. The guide cannulas were inserted using the following coordinates from the bregma: 0.5 mm anterior, 2.5 mm lateral, and 2.7 mm vertical. Seven days post-surgery, while the animals were awake, they received injections of haloperidol or saline solution. This involved lowering the injection cannula connected to a micro-infusion pump through the guide cannula into the animal’s basal ganglia. Aminophylline or saline solution (1 μL per brain hemisphere) was then administered at a rate of 0.5 μL/min. After completing the drug infusion, a 1-min pause was observed, the injection cannula was removed, and the animals underwent catalepsy and gait tests. Figure 5 illustrates the experiment. 

### 4.2. Assessment of Catalepsy and Motor Behavior

The assessments were performed using the catalepsy test [55,56,57] and the Open Field Maze [58,59,60]. The tests were conducted at the Laboratory of Biopharmacology of the Federal University of Mato Grosso do Sul (UFMS, Brazil) at the same time of day (from 1:00 p.m. to 4:00 p.m.), in the same testing room, and by the same researcher. 

The catalepsy test involves placing an animal in an unusual posture and recording the time taken to correct this posture. In this study, catalepsy was evaluated according to the bar method, in which the mouse was positioned with both front paws on a horizontal metal bar (diameter of 0.5 cm) elevated 4.5 cm from the ground (Figure 6a). The time in seconds during which the animal remained in this position was recorded, allowing three attempts to place the animal in a cataleptic position. The catalepsy time was considered to be complete when the front paws touched the ground or when the mouse climbed onto the bar. Measurements were taken 5, 35, and 65 min after the intracerebral administration of aminophylline or saline.

The Open Field Maze is one of the most commonly used platforms for measuring motor behavior in animal models. It is a fast and easy test that provides a variety of information regarding motor functions. In this study, the animal was positioned in the central region of a cylindrical arena with translucent acrylic walls (40 cm in diameter and 30 cm in height) that were placed on a white base divided into 12 quadrants. The test was performed 30 min after removal of the intracerebral injection cannula using aminophylline or saline solution. The parameters evaluated during the Open Field Maze were walking distance (number of quadrants crossed, Figure 6b) and vertical exploration (number of climbs, Figure 6c). 

### 4.3. Animal Disposal

After the experiment, the animals were anesthetized with ketamine (100 mg/kg) and xylazine (10 mg/kg) at an application volume of 10 mL/kg. The animals were euthanized by cervical dislocation and sent for incineration.

### 4.4. Statistical Analysis

Statistical analysis was performed using SigmaPlot software 10 (Systat Software, Inc., San Jose, CA, USA). First, the data were checked for normality (Shapiro–Wilk test) and the homogeneity of variances (Levene test). Confirming the assumptions necessary for performing the parametric tests, we proceeded with proper statistical procedures.

Data are reported as mean ± standard deviation. A two-way repeated-measures analysis of variance (ANOVA), followed by Tukey’s post hoc test, was employed to evaluate the impacts of group (experimental × control), aminophylline (60 nM × 120 nM × saline/placebo), and interactions on catalepsy (measured by the time during which the animal remained on the bar). For gait (quadrants crossed) and vertical exploration (number of climbs) in the Open Field Maze, a one-way ANOVA was performed, followed by Tukey’s post hoc test. Significance was set at 5% [61,62].

## 5. Conclusions

Haloperidol induced catalepsy and significantly affected horizontal and vertical exploration by the animals, replicating signs similar to those of PD. The cataleptic effect was reversed by the intracerebral administration of aminophylline. In contrast, aminophylline was not effective in reversing hypokinetic symptoms. These findings suggest that additional doses of aminophylline associated with other substances should be tested to reverse haloperidol-induced motor dysfunction. 

Our study highlights important adenosinergic interactions in animals with and without signs comparable to those in PD. These findings offer valuable insights into the neurobiology of PD and emphasize the importance of exploring novel therapeutic strategies for improving catalepsy and gait. Further studies should explore the effect of other doses of aminophylline, combined with or without other substances, in order to improve motor functions. 

## Figures and Tables

**Figure 1 ijms-25-05191-f001:**
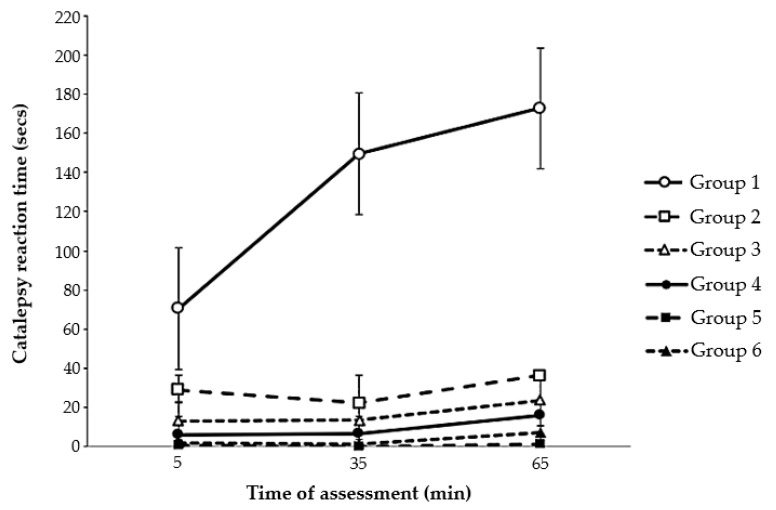
Results of catalepsy reaction time. Note: Group 1: PD-induced model with saline solution; Group 2: PD-induced model with aminophylline at 60 nM; Group 3: PD-induced model with aminophylline at 120 nM; Group 4: Non-PD-induced model with saline solution; Group 5: Non-PD-induced model with aminophylline at 60 nM; Group 6: Non-PD-induced model with aminophylline at 120 nM.

**Figure 2 ijms-25-05191-f002:**
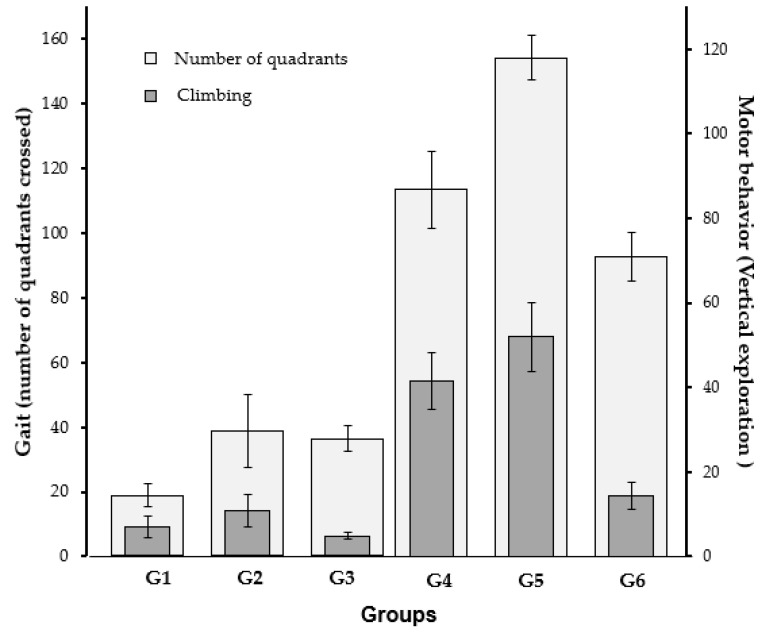
Results of Open Field Maze.

**Figure 3 ijms-25-05191-f003:**
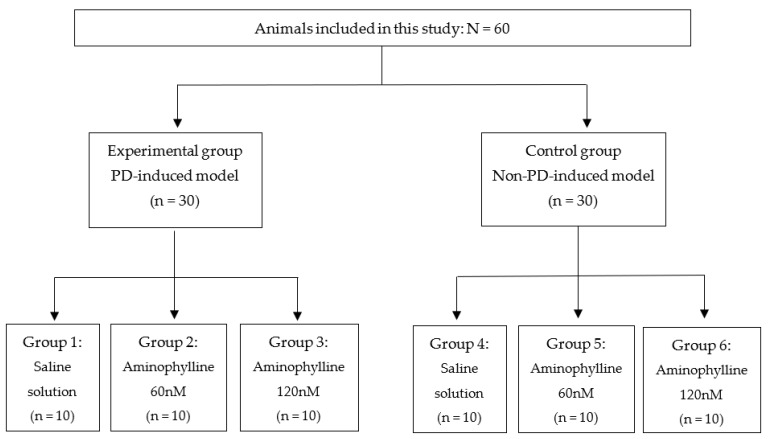
Division of animals into the experimental and control groups.

**Figure 4 ijms-25-05191-f004:**
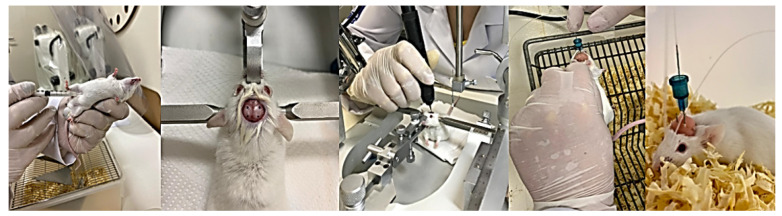
Surgical procedures adopted in this study.

**Figure 5 ijms-25-05191-f005:**
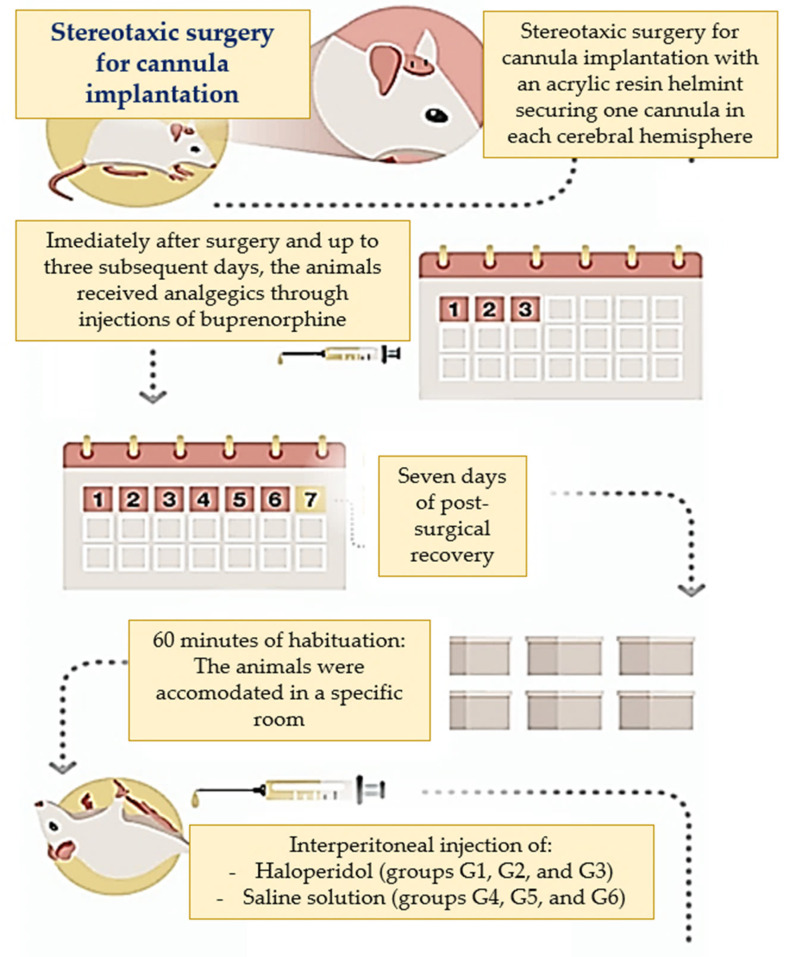
Illustrative scheme of the experiment.

**Figure 6 ijms-25-05191-f006:**
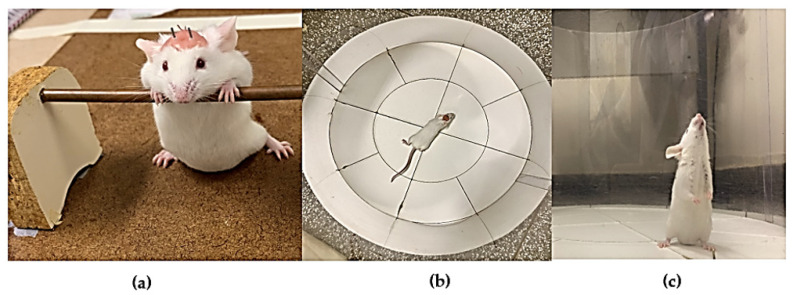
Catalepsy test (**a**) and the Open Field test (**b**,**c**).

## Data Availability

The data presented in this study are available on request from the corresponding author.

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
