# Peer review of "Effects of Intracerebral Aminophylline Dosing on Catalepsy and Gait in an Animal Model of Parkinson’s Disease"

_ijms, 2024, doi:10.3390/ijms25105191_

Round 1

Reviewer 1 Report

Comments and Suggestions for Authors

In this work, the authors performed pharmacological and behavioral activity assessment in laboratory mammals in order to study the effect of aminophilin on cataleptic symptoms and behavioral activity, in particular the ability to move when modeling PD using haloperidol.

During the review, the following comments on the work arose:

1. The Introduction section is quite complete and provides a justification for the need for these studies. The authors fully and in sufficient detail described the need for these studies to expand approaches to pharmacological correction of PD, in particular motor symptoms.

2. Perhaps the authors should more clearly present the stages of experimental exposure to animals. Despite the presented scheme of the experiment, the experimental part needs a more visual presentation, these may be fragments of video and photos containing the stages of administration of haloperidol, aminophylline and, in particular, a demonstration of behavioral and motor tests in an open field.  The photographs of animals presented by the authors do not give an idea of the experiments carried out. 

3. In the Lemitation section, the authors quite correctly indicate the need to examine animals before and after experimental manipulations. This is an absolutely necessary step that can prevent the non-obvious side effects of aminophilin administration. The authors are strongly encouraged to supplement their research with these data.

Comments on the Quality of English Language

English language correction is required

Reviewer 2 Report

Comments and Suggestions for Authors

The manuscript "Effects of Intracerebral Aminophylline Dosing on Catalepsy and Gait in an Animal Model of Parkinson’s Disease" explores the therapeutic potential of aminophylline in alleviating Parkinson's disease (PD) symptoms in mice. The study used sixty Swiss mice, divided into control and experimental groups, with the latter induced with catalepsy via haloperidol. Aminophylline was administered at two concentrations (60nM and 120nM) to test its efficacy in reversing catalepsy and improving gait, assessed using the bar test and Open Field Maze.

Results showed that aminophylline effectively reversed catalepsy at both doses but did not improve gait disturbances. This outcome suggests a dose-dependent effect of aminophylline, which was stimulating at 60nM but not at 120nM in control animals. The manuscript discusses these findings in the broader context of PD treatment, noting the limitations of current therapies like levodopa and the potential of targeting non-dopaminergic pathways.

In conclusion, the study underscores aminophylline's partial efficacy in treating PD symptoms, highlighting the need for further research into dose optimization and combination therapies. The findings contribute to the growing interest in alternative PD treatments, emphasizing the necessity of comprehensive future studies to explore underlying mechanisms and therapeutic potential.

Introduction

1. The introduction of Parkinson's disease (PD) is clear, but it would be beneficial to briefly explain why these particular symptoms (bradykinesia, muscle rigidity, etc.) are significant to the pathology of PD, which could help non-specialist readers understand the severity and impact of these symptoms. Consider adding a sentence or two about the general prevalence of PD to give readers a sense of its impact globally.

2. In the discussion of neurotransmitter systems affected by PD, it could be helpful to specify how these dysfunctions contribute to the symptoms or progression of the disease. This could involve a short explanation linking specific neurotransmitters to particular PD symptoms or disease mechanisms.

3. The section on adenosinergic pathways is quite detailed, but it might benefit from a simplified summary or a figure that illustrates the interactions between these pathways and the dopaminergic system in the context of PD.

4. Ensure that all studies mentioned (e.g., those referring to the effectiveness of levodopa and the role of adenosine in the basal ganglia) are cited correctly and that the most current and relevant research is included. This could involve checking that recent seminal papers have not been overlooked. The manuscript frequently cites ranges of studies (e.g., [12-14], [19-22]); it would enhance credibility to directly reference specific studies that strongly support the claims made, particularly for critical points.

5. The phrase "Bering on mind that numerous pathways are affected in PD" seems to contain a typo and could be corrected to "Bearing in mind that numerous pathways are affected in PD." Consider revising complex sentences to improve readability and flow, particularly in the sections discussing biochemical pathways and receptor interactions.

6. Discussion on the limitations of current therapeutic strategies, particularly those involving levodopa, could be expanded. This could include more details on the nature of these limitations (e.g., side effects or reduced efficacy over time) and how they impact treatment decisions.

7. The manuscript could benefit from a brief discussion of ongoing clinical trials or emerging therapeutic approaches targeting non-dopaminergic systems, providing a broader perspective on the future of PD treatment.

8. Highlight the potential for new discoveries in adenosinergic agents and other areas as crucial for developing more effective and sustainable PD treatments.

Results

9. The description of the experimental setup and the results is clear, but it could benefit from a bit more detail on the specifics of the catalepsy test and open field maze test. For instance, how was catalepsy quantified? What specific behaviors were measured in the open field maze?

10. The statistical outcomes are well-presented, including the significance levels and the effects observed. However, the manuscript could include more specific details about the statistical methodology, such as the assumptions checked for ANOVA, which would help in reinforcing the validity of the statistical conclusions.

11. Ensure consistency in reporting units and treatment descriptions across the section. For example, the manuscript switches between "aminophylline" and "dose" without specifying the unit consistently. It’s mentioned as "60nM" and "120nM" without establishing these are consistent or the reasons for choosing these specific doses.

12. The interpretation of the findings could be expanded. For example, while it is noted that aminophylline improved catalepsy symptoms at both tested doses, the discussion on why the 60nM dose was more effective in the open field test could be elaborated. This might include hypotheses about dose-response relationships or the pharmacokinetics of aminophylline.

13. There's a lack of discussion about why aminophylline did not improve vertical exploration, which could be significant for understanding the limitations of this treatment in addressing motor symptoms in PD.

14. It would be beneficial if Figure 4 (catalepsy reaction times) and Figure 5 (results of Open Field Maze) could include each group or confidence intervals to help readers assess the variability within groups.

15. A table summarizing the key findings from each group, including mean values, standard deviations, and p-values, could help readers quickly understand the data.

16. Briefly comparing these results with other relevant studies could provide context and show where this study's findings fit within the broader research landscape. This might include comparing the effectiveness of aminophylline to other treatments or discussing any novel observations made in the study.

17. Suggest potential future research based on these results. For instance, exploring other doses or combinations with other therapies might be worthwhile, especially since aminophylline did not improve all aspects of motor dysfunction in the PD model.

18. There are minor grammatical issues and potential typographical errors to correct, such as "aminophylline did not show any benefits compared with those in group 4" could be clearer. Also, ensure consistent formatting for P-values (e.g., P < 0.05, not P < 0,05).

Discussion

19. It's good to see a strong linkage between the results presented and the discussion points. However, ensure that all claims made in the discussion can be directly tied back to the data presented in the results section. For instance, the claim about aminophylline acting as a stimulant at certain doses should be substantiated with specific results.

20. The comparison with previous studies is well handled, showing how your findings align or differ from existing research. To strengthen this section, you might consider discussing possible reasons for any discrepancies between your results and those of others, such as differences in methodology, animal models, or dosages. It would be beneficial to discuss the broader implications of these findings for the field of PD treatment, particularly in the context of non-dopaminergic therapies.

21. The discussion about the interactions between adenosine and dopamine receptors is insightful. Expanding on how these findings might influence future drug development or lead to novel therapeutic strategies could be very valuable. Consider discussing the potential for targeting these pathways in human clinical trials based on animal model findings.

22. The limitations section is essential and well-placed. You might want to expand on this by discussing additional limitations related to the study design, such as the potential effects of chronic versus acute treatment or the generalizability of the findings from mice to humans.

23. While you mention that other doses and substances should be tested, it could be useful to provide specific recommendations or hypotheses based on the current findings. For example, suggest potential combinations of aminophylline with other pharmacological agents that could be explored, or propose further studies to investigate the dose-response relationship in more detail.

24. Ensure consistency in terminology and detail when discussing specific receptors and drugs. For instance, when mentioning receptor subtypes, consistently specify whether they are adenosine or dopamine receptors to avoid any confusion.

25. Some sentences could be rephrased for clarity and impact. For example, instead of saying "This result suggests a greater efficacy of aminophylline," you could say, "These results suggest that aminophylline may be more efficacious, potentially due to its mode of administration."

26. Consider whether additional figures or conceptual diagrams could help illustrate the complex interactions discussed, such as the interplay between adenosinergic and dopaminergic systems in the basal ganglia.

Comments on the Quality of English Language

Some sentences are overly complex or verbose, which may hinder readability and understanding. It would be beneficial to simplify the language where possible, breaking down complex ideas into shorter, clearer sentences.

Round 2

Reviewer 1 Report

Comments and Suggestions for Authors

Dear authors, you have made changes to the manuscript and improved the article by adding links and expanding the sections Introduction, Materials and Methods, and Discussion. However, the main comments, the primary reviews of paragraphs 2 and 3 were ignored. In accordance with the "Data Availability Statement" you have indicated, in particular, fragments of videos and photographs containing the stages of administration of haloperidol, aminophylline and, in particular, a demonstration of behavioral and motor tests in an open field should be provided in accordance with the reviewer's request. Further, in the section "Limitations", the authors quite correctly point out the need to examine animals before and after experimental manipulations. This is a necessary step that can prevent non-obvious side effects from the administration of aminophylline. The authors are recommended to supplement their research with these data. The authors' explanations that they will provide relevant data in the next publication are unsatisfactory.

Comments on the Quality of English Language

English is satisfactory

Reviewer 2 Report

Comments and Suggestions for Authors

I want to sincerely thank the authors for their thorough and considerate revisions. The manuscript has significantly improved from its original submission, clearly reflecting the authors' dedication to addressing the previously raised concerns.

Comments on the Quality of English Language

Minor editing of English language required

Round 3

Reviewer 1 Report

Comments and Suggestions for Authors

The necessary changes have been made to the work, satisfying the previously expressed comments. In its present form, the work can be published in the IJMS